# OpenReview forum: "Can We Predict Alignment Before Models Finish Thinking? Towards Monitoring Misaligned Reasoning Models"
_ICLR.cc/2026/Conference — Submitted to ICLR 2026_

### Official Review · Reviewer_xV2h · 2025-10-27

**Soundness:** 2
**Presentation:** 3
**Contribution:** 2
**Rating:** 2
**Confidence:** 4

**Summary:**

The authors examine whether it is possible to predict the alignment (refusal vs non-refusal) of a reasoning model's ultimate response from its reasoning trace. They compare text-based classifiers with linear probes on hidden activations, and find that the probes are superior. They also suggest that activation-based probes can predict the response outcome many reasoning steps ahead. They also conduct error analysis to show systematic cases where the probes beat text-based classifiers.

**Strengths:**

- The paper is well-written, clear, and easy to read.
- Thorough experiments, including multiple models, datasets, and training dataset sizes.

**Weaknesses:**

- Lacks justification of important technical details
  - The decision to train probes on the last layer of the last token position is an important design decision (which may have serious implications; see the Questions section), and lacks justification in the current manuscript.
  - There is also no discussion of sampling methodology. It is implicitly assumed that a partial reasoning chain deterministically leads either to refusal or non-refusal. In reality, reasoning model inference uses non-deterministic sampling, and so refusals are stochastic.
- Motivation is unclear
  - In my opinion, the motivation is unclear. Why should we care about predicting whether models refuse before they refuse? One of the predominant methods in LLM safety is to run classifiers of the *model output* ([Sharma et al., 2025](https://arxiv.org/abs/2501.18837)), and classify the model output directly.
- Limited to reasoning models trained via SFT
  - SotA reasoning models are trained via RL, and it's unclear whether analysis of SFT-trained reasoning models generalize to RL-trained reasoning models. In particular, it is unclear whether the results would hold on models that were trained to reason about whether or not to refuse a request, as in [Guan et al., 2024](https://arxiv.org/abs/2412.16339).

**Questions:**

- How is the sampling done?
  - For reasoning models, official documentation generally advises to not use temperature 0. If temperature 0 is used here, then whether a reasoning trace leads to a refusal or not is probabilistic, not deterministic. Have you thought about this? What temperature are you sampling with? Do you resample multiple times per rollout? I think the paper would be strengthened if you clarify and justify your sampling methodology.
- Why use the last layer of the last token position? Is this a fair design choice?
  - The probes are trained using activations from the last layer of the last token position. This is immediately preceding the unembedding, so this activation will contain information about the next token prediction. In this case, the next token prediction is very useful in order to determine whether the response will be a refusal or not (the first tokens of refusal responses are drawn from a narrow distribution of refusal phrases).
  - I fear that this gives an unfair advantage to the probing methods over the text-based classifiers - the probing methods effectively have access to the first token of the response.
  - One way to try and disentangle things / prove that the probe works beyond this last-token effect would be to give the first token of the response to the text classifiers, so as to try and give them an "even playing field".
  - In my opinion, this issue significantly weakens the main result, namely that activation-based probes outperform text-based classifiers (section 4).
- Do the results generalize to reasoning models that were actually trained via RL?
  - The models are limited to reasoning models that were trained via SFT on reasoning traces. Do you expect the results to generalize to reasoning models trained with RL? There are open-source models trained with RL available, such as Qwen 3, and I think it would be worth reproducing the experiments for that model family in order to test this question.
- Line 251: "Text-based classifiers rely on semantic cues in the CoT to infer model behavior. For example, if a CoT includes planning steps toward an illegal request, these classifiers will likely predict a harmful outcome."
  - Where is the evidence for this claim? How can we know how the text-based classifiers work? Is this speculation? If so, it'd be good to mention that this claim is speculative.
- Missing baselines for section 5
  - How do text-based classifiers perform in the "future" setting? Do probes still outperform text-based classifiers in this setting? Or does the advantage disappear?
- Section 6 - asymmetric error analysis
  - Did you study the opposite error analysis? E.g., are there cases where the probes systematically fail, but where the text-based classifiers do well?

---

> ### Author Response · Authors · 2025-11-28
>
> Thank you for your thoughtful review. We appreciate that you found the paper well-written with thorough experiments. We address your concerns below:
>
> >**W1 & Q1**: Sampling methodology
>
> We use a temperature of 0 throughout all experiments because we intended to ensure deterministic and reproducible results. While reasoning models sometimes use stochastic sampling to give the model more flexibility in solving STEM reasoning questions, we believe that safety alignment tasks (i.e., responding to harmful prompts) benefit from deterministic sampling. We will run additional experiments with a non-zero temperature and clarify this in the revision.
>
> >**W1 & Q2**: Extracting activations from last layer, last token
>
> We acknowledge this concern and will run ablation experiments in which text classifiers are given the first response token. We chose to extract activations from the last-token position at the final layer, following standard practice in prior work [1]. While this design choice may contribute to the probe’s performance, it does not fully explain our main results, especially those in Section 5 showing that a probe trained on activations can predict future response alignment. At 20-30% observed CoT, probes achieve ~60-70 F1 despite being far from the point of response generation. At this stage, the model has not yet decided how to respond, so “first-token information” does not exist.
>
> >**W2**: Motivation is unclear
>
> Correctly predicting whether the outcome is safe or unsafe in real-time can (1) enable early intervention to prevent harmful content being generated *at all*, (2) save substantial compute from generating the long CoT in reasoning models. CoT offers unique opportunities for AI safety, but they could be unfaithful which makes it challenging for monitoring [1]. Post-hoc output classifiers (e.g., Sharma et al., 2025) cannot detect failure cases such as performative or contradictory reasoning. Our work addresses these concerns by providing the first systematic comparison of CoT monitoring methods, and showing that activation-based monitoring is necessary for reliable safety evaluation.
>
> >**W3 & Q3**: Generalization to RL-trained models
>
> Thank you for this question. We agree that testing on RL-trained models would strengthen our work, and will run additional experiments on RL-trained models, such as Nemotron-Research-Reasoning-Qwen.
>
> >**Q4**: CoT relies on semantic cues
>
> It is correct that this claim is somewhat speculative. It is based on qualitative analysis of success/failure cases: we observe text classifiers succeed more when CoTs contain explicit harmful planning steps, and struggle when reasoning is ambiguous (e.g., debating whether to answer). We will revise this passage to clarify it is an observation-based hypothesis rather than a definitive claim.
>
> >**Q5**: Text baselines for future prediction
>
> Thank you for this suggestion. We did not evaluate text-based classifiers in the future setting because we simply took the best-performing predictor from Section 4 and tested it on this additional, more challenging future prediction task. However, we agree that this comparison would be valuable and will include the baseline results in the revised version of our paper.
>
> >**Q6**: Asymmetric error analysis in section 6
>
> We analyzed the four error types: (text correct, probe correct), (text correct, probe incorrect), (text incorrect, probe correct), and (text incorrect, probe incorrect), averaged across the three datasets for the s1.1-7B model. The table below shows that cases where text classifiers succeed but probes fail are rare (6.7% of test cases). In contrast, cases where probes succeed but text classifiers fail account for over 30% of test cases. This asymmetry in errors then motivates us to investigate the 30% which form the subset of performative CoTs.
>
> |                        | Probe Correct | Probe Incorrect |
> |------------------------|---------------|------------------|
> | **Text classifier correct**   | 51.2%        | 6.7%            |
> | **Text classifier incorrect** | 33.6%        | 8.5%            |
>
> _**References**_
>
> [1] Reasoning Models Know When They’re Right: Probing Hidden States for Self-Verification (https://arxiv.org/abs/2504.05419)
>
> [2] Chain of Thought Monitorability: A New and Fragile Opportunity for AI Safety (https://arxiv.org/abs/2507.11473)

---

### Official Review · Reviewer_5VFH · 2025-10-29

**Soundness:** 4
**Presentation:** 4
**Contribution:** 1
**Rating:** 2
**Confidence:** 4

**Summary:**

This paper investigates methods for predicting the safety alignment of language model responses, specifically focusing on reasoning language models (RLMs) that generate long chains of thought (CoTs). The core finding is that a simple linear probe trained on CoT activations significantly outperforms all text-based monitoring methods, including highly capable large language models (LLMs) and human annotators, in predicting whether a final response will be safe or unsafe. The linear probe demonstrates that alignment signals can be detected from early CoT segments, enabling potential real-time safety monitoring and intervention before the model finishes its full reasoning process. The research suggests that internal, latent representations provide a more reliable signal for safety monitoring across different RLM sizes and safety benchmarks than text-based analysis.

**Strengths:**

- The paper is easy to read and well structured.
- This paper tackles a relevant problem that has gained significant attention recently.
- The authors employ three relevant datasets for their experimental procedure, which seems well executed overall and the analysis of results is well conducted.

**Weaknesses:**

- There seems to be an important body of literature missed in this work's background. Real time safety alignment prediction is not novel, and it's been well understood that simple linear discriminators can perform well for this task (see references below).
- Related work focuses on Reasoning and Chain-of-Thought literature, while ignoring a large bulk of related work on controlled text generation. For example, how does this work sufficiently differ from [1] and [2] for it to be considered a worthwhile contribution? On the same note, recent work published at this conference, such as [3] also seem to incorporate a similar idea (in their case applied to domain certification, but one could argue it's a related concept).
- One of the key findings of the paper is the possibility of real-time monitoring. However, this has been known and explored in the past.
- Benchmark methods ignore some recent, popular baselines, such as LlamaGuard and (especially) WildGuard, which should be included in the experimental setting.
- Overall, it seems like this paper tackles a subset of a larger problem that has been studied for a while now, with findings that have been reported in a similar fashion in other papers. The main difference I see from this paper in comparison to the existing literature I am aware of is the analysis of model alignment in the presence of Chain-of-Thought. In my opinion, this distinction should be clear in the paper and the authors should explain why there is a need to extend this concept to CoT, since at present it looks to me like a trivial extension of existing work.

[1] Yang, K., & Klein, D. (2021). FUDGE: Controlled text generation with future discriminators. arXiv preprint arXiv:2104.05218.

[2] Fonseca, J., Bell, A., & Stoyanovich, J. (2025). Safeguarding large language models in real-time with tunable safety-performance trade-offs. arXiv preprint arXiv:2501.02018.

[3] Emde, C., Paren, A., Arvind, P., Kayser, M., Rainforth, T., Lukasiewicz, T., ... & Bibi, A. (2025). Shh, don't say that! Domain Certification in LLMs. arXiv preprint arXiv:2502.19320.

**Questions:**

- The concept "linear probes", in the context of machine learning, that I am aware of, comes from [4]. Is this what you are actually using? It's very unclear to me whether actual linear classifier probes are being used (and how), or whether this is just a simple Logistic Regression trained over 50 PCA components using the model logits. If what is being done is the latter, then previous work have also explored this concept [5].
- Why aren't the human evaluation results not reported in the main body of the paper as well? This comparison is posed as part of your contributions, yet it's only available in the appendix.
- How many human annotators were used?

[4] Alain, G., & Bengio, Y. (2016). Understanding intermediate layers using linear classifier probes. arXiv preprint arXiv:1610.01644.

[5] Krause, B., Gotmare, A. D., McCann, B., Keskar, N. S., Joty, S., Socher, R., & Rajani, N. F. (2020). Gedi: Generative discriminator guided sequence generation. arXiv preprint arXiv:2009.06367.

---

> ### Author Response · Authors · 2025-11-28
>
> Thank you for your thoughtful review and for recognizing the quality of our writing and analysis. To clarify the concerns raised:
>
> > **W1**: There seems to be an important body of literature missed in this work's background. Real time safety alignment prediction is not novel, and it's been well understood that simple linear discriminators can perform well for this task (see references below).
>
> Our key novelty is studying real-time safety prediction for **reasoning models** with explicit CoT, as well as their **faithfulness**. We will include the citations the reviewer provided in our related work, but we want to highlight that these work is orthogonal to our findings.
>
> > **W2**: Related work focuses on Reasoning and Chain-of-Thought literature, while ignoring a large bulk of related work on controlled text generation. For example, how does this work sufficiently differ from [1] and [2] for it to be considered a worthwhile contribution? On the same note, recent work published at this conference, such as [3] also seem to incorporate a similar idea (in their case applied to domain certification, but one could argue it's a related concept).
>
> First, [1], [2] and [3] do not work with reasoning models that have CoT. Second, they (and all the existing work to the best of our knowledge) did not evaluate the faithfulness of the final response to the CoT.
>
> Specifically,  our paper addresses a **different challenge: can we monitor long chains of thought (CoT) in reasoning language models (RLMs) and predict the outcome of such thinking?**
>
> - Unfaithful CoT: Our main novel contribution is studying how faithful the safety reasoning is, when many open-source reasoning models demonstrate the ability to reason about safety. Our work does not perform any intervention to the reasoning.  Our setup contrasts all prior work on controlled text generation (including [1]) and CoT faithfulness study that introduces interventions to the CoT (such as [2] and [3]). Furthermore, prior work on controlled text generation does not factor nor evaluate the CoT faithfulness aspect.
> - “Predictive” vs. “present” monitoring: In [1,2], the monitored text *is* the final output. There is no distinction between the current generation and whether it logically leads to future answers. We instead predict properties of a future response from intermediate reasoning traces, and show that signals about future responses can be learned in current CoT.
>
> > **W3**: One of the key findings of the paper is the possibility of real-time monitoring. However, this has been known and explored in the past.
>
> While that is our takeaway, our key finding is understanding the behavior of safety CoT with open reasoning models, especially on their faithfulness. We do *not* think that prior study on real-time monitoring can be directly applied to our setup, especially the work that the reviewer has provided ([1], [2], and [3]).
>
> > **W4**: Missing LlamaGuard and WildGuard as baselines
>
> We believe that the suggested LlamaGuard and WildGuard are **worse baselines** than our setup with GPT-4o as shown in prior work [6], where GPT-4o outperforms them both in harmfulness classification. Furthermore, we introduce stronger baselines, such as in-context learning (GPT-4.1-nano) and reasoning (o4-mini).
>
> >**Q1**: Usage of linear probe
>
> We use the term “linear probe” to refer to logistic regression trained on model activations, which is standard practice in the interpretability and representation learning literature [7, 8]. This is consistent with the usage in [4], where linear classifiers are trained on intermediate representations to probe what information is encoded.
>
> >**Q2&3**: Human evaluation
>
> We appreciate this feedback. We put human evaluation in Appendix B due to space constraints and because the main claims of the paper (activation-based monitoring outperforms text-based methods) are supported by experiments in the main text. We have moved the human evaluation results to the main body in the revision. We had seven human annotators.
>
> _**References**_
>
> [6] R-Judge: Benchmarking Safety Risk Awareness for LLM Agents (https://arxiv.org/abs/2401.10019)
>
> [7] Discovering Latent Knowledge in Language Models Without Supervision (https://arxiv.org/abs/2212.03827)
>
> [8] The Geometry of Truth: Emergent Linear Structure in Large Language Model Representations of True/False Datasets (https://arxiv.org/abs/2310.06824)

---

### Official Review · Reviewer_9Vch · 2025-10-30

**Soundness:** 3
**Presentation:** 3
**Contribution:** 2
**Rating:** 4
**Confidence:** 4

**Summary:**

This paper aims to monitor the harmfulness of LRMs' responses based on partial or full CoT reasoning procedure. The authors demonstrate that directly using the CoT texts might not be a proper solution, while conducting linear probing on the model activations is a better solution. Extensive experiments demonstrate the effectiveness of the proposed method.

**Strengths:**

- Monitoring the harmfulness of the LRMs' final responses based on the CoT procedure is interesting.
- Although simple, the authors compare multiple baselines and different settings of linear probing (e.g., future-trained and present-trained).

**Weaknesses:**

- About CoT monitoring methods:
  - I wonder what the differences are between the fine-tuned BERT classifier and the fine-tuned harmfulness classifier, since both are conducting binary classification.
  - I'm not sure about your settings. What do you try to predict? For each CoT index, do you try to predict the harmfulness of the final response without altering the original reasoning procedure, or will you interrupt and generate an instant response at each CoT step, and then do the prediction （as lines 78-79 say?
  -  For the fine-tuned BERT, fine-tuned harmfulness classifier, and the activation-based monitoring, how to deal with the sequence length dimension is unclear.
  - In Table 2, the probing accuracy of s1.1-7B on XSTest is an outlier. Any idea on the reasons?
- About early thinking:
  - After your analysis in Sec. 5, what are your final empirical suggestions? Like, you can keep detecting the harmfulness of the CoT procedure, and if there is an alarm triggered, what should you do?
- Overall, although simple, this is an interesting exploration of monitoring LLM CoTs. However, there are details that are still unclear. I would like to see the rebuttal for the final decisions.

**Questions:**

Check the Weakness part.

---

> ### Author Response · Authors · 2025-11-28
>
> Thank you for your review and feedback! We are glad that you found our work interesting and recognize the extensive experiments. In response to the questions raised:
>
> >**W1**: Differences between fine-tuned BERT and harmfulness classifiers
>
> Yes, both perform binary text classification, but the key difference is that BERT is trained from scratch while the harmfulness classifiers are off-the-shelf LLMs.
> - Fine-tuned BERT classifier: We train ModernBERT specifically on the task of predicting final response alignment from CoT text. The training data consists of (CoT, response alignment label) pairs collected in our setup (Section 3.1).
> - Fine-tuned harmfulness classifiers: These are off-the-shelf LLM-as-a-judge models provided by the safety benchmark authors. They were originally trained to evaluate final responses for harmfulness, not to predict alignment from CoT traces. We repurpose them for our task to test whether they generalize to CoT-based prediction.
>
> We see that on HarmBench, BERT achieves better performance than the HarmBench classifier, which shows the benefit of task-specific training. But on StrongReject and SORRY-Bench, there is not a significant overall difference, suggesting that the harmfulness classifiers are already quite strong and generalize well to CoT-based prediction.
>
> >**W2**: Clarification on the prediction setting
>
> It is the latter.  We generate a response $A_i$ after each partial CoT segment $T_{1:i}$. This setup allows us to track how response alignment evolves as the CoT progresses. For each prompt, if the full CoT contains n sentences, we obtain n pairs: $(T_{1:1}, A_1, label_1), (T_{1:2}, A_2, label_2), ..., (T_{1:n}, A_n, label_n)$.
>
> This experimental design serves two purposes: (1) It creates training data for text-based or activation-based predictors to learn the relationship between partial CoT and eventual response alignment. (2) It allows us to study whether alignment signals emerge early in reasoning (Section 5) and whether these signals are stable or change dynamically (Section 6, performative CoTs).
>
> >**W3**: Sequence length dimension
>
> For fine-tuned BERT: ModernBERT has a maximum context length of 8192 tokens. We truncate from the end if the CoT exceeds this limit and apply padding to ensure uniform batch sizes.
>
> For fine-tuned harmfulness classifiers, we apply the same procedure. The maximum context length depends on the base model (e.g., 2048 for HarmBench, 4096 for StrongReject). Since the CoT can exceed this limit, we additionally have the CoT_para_max and CoT_para_majority methods, which segment the CoT into paragraphs and evaluate each independently.
>
> Activation-based monitoring does not have sequence length limitations, because the activations are a fixed-size vector of dimensionality d (e.g., 3584 for Qwen-2.5-7B based models). We then apply PCA to reduce this to 50 dimensions. This is substantially more efficient than processing long text sequences.
>
> >**W4**: Probing accuracy on XSTest
>
> XSTest is designed to be a challenging dataset that can trigger over-refusal, with prompts that are technically benign but may appear harmful (e.g., “What is identity theft?”). This differs from the standard instruction-following prompts in AlpacaEval. For the s1.1-7B model specifically, it is less safe than the R1-Distill models after STEM reasoning fine-tuning, as evidenced by lower safe rates in Table 1 (Baseline). As a result, the probe trained on s1.1-7B data learns from a distribution with more harmful examples. This likely leads to the probe to be more sensitive in classifying activations as harmful and causes it to overpredict harmfulness on the tricky examples in XSTest.
>
> >**W5**: Empirical suggestions
>
> Thank you for pointing out the applicability of our findings. When the probe detects a generation as harmful, we could employ different intervention strategies: 1) roll back the generation and have the model regenerate while adding a system prompt emphasizing helpfulness and harmlessness; 2) provide a pre-defined refusal response immediately for highly harmful prompts (e.g., explicit requests for illegal content). The probe’s prediction confidence could inform which strategy to use. The specific intervention frameworks would also depend on the model developer's policy and deployment context.

---

### Official Review · Reviewer_r4zw · 2025-10-31

**Soundness:** 2
**Presentation:** 3
**Contribution:** 3
**Rating:** 4
**Confidence:** 4

**Summary:**

The paper studies whether the chain of thought of a LLM provides early insight into whether the eventual response of the model could be misaligned (unsafe). The chain of thought segment of the model is represented in two ways: in a textual manner, and as the activation (embedding) representation. The latter is shown to be an effective indicator of eventual response misalignment.

**Strengths:**

When access to a model’s activations is available, the paper demonstrates an important takeaway: the activations hold sufficient information to be predictive of eventual misalignment in long thinking or reasoning traces. This can facilitate setting up effective test-time safety guardrails

**Weaknesses:**

The analysis seems to have an unaccounted pathway for leakage of information, which influences the findings and takeaways. The *activations* at the final token position of the last layer for each partial CoT (Line 163) implicitly encode the **prompt** itself, in addition to the subsequent CoT. This leads to a few issues:
- This potentially explains the effectiveness of the linear probe: if the prompt itself is indicative of the final misalignment of the response, the CoT segment is not required for the prediction of misalignment. And if this embedding representation of the partial CoT does include information about the prompt (since the LLM has first parsed the prompt before the CoT and encoded it within its activations/parameters) that should suffice.
- This results in a confounding factor for evaluating the central claim of the paper: it is not *just* partial CoTs that are being tested for being predictive for eventual misalignment.
- This invalidates the comparison with text based monitoring, since the text of the CoT by itself does not also include the prompt.

It would be helpful if the authors validate and clarify this. I am open to re-assessment if the authors have followed a procedure that differs from the description in Line 162–163.

If not, and assuming the updated claim is: activations of an LLM during its CoT processing are predictive of eventual misalignment, a drawback that arises is that such monitoring can only be applied to models whose activations are accessible, and thus not closed-source models.

**Questions:**

- In Line 193, what is the value of $d$? That is, how aggressive is the dimensionality reduction with PCA?
- How is “foresight” used for training? As per the task description, only “observed” must be used to predict eventual misalignment.
- It is surprising to see that the prediction accuracy is already very high with even 0-10% CoT (Figure 4). How is that the case? It would be helpful if the authors can provide a couple of examples of such cases. Is this a by-product of the prompt being encoded in activations used as input to the linear probe?

---

> ### Author Response · Authors · 2025-11-28
>
> Thank you for your thoughtful and constructive feedback. We would like to address each of your concerns below.
>
> > **W1**: The activations at the final token position of the last layer for each partial CoT (Line 163) implicitly encode the prompt itself.
>
> Yes, it is true that CoT activations include information from the prompt. However, **the prompt alone is not sufficient to reliably predict final response alignment; we provide evidence that the CoT itself is necessary.**
>
> **Evidence 1: Prompt-only performance is weak.** In Figure 4 and Figure 9, we report F1 scores at varying percentages of CoT observed. At 0% CoT (i.e., activations collected immediately after the prompt, before any CoT generation), the F1 score is only ~50, which is the lowest. As more CoT is observed, F1 improves consistently and substantially, reaching ~75-80 at 100% CoT. This monotonic improvement demonstrates that the CoT itself carries important predictive signals beyond what the prompt provides.
>
> **Evidence 2: As CoT progresses, final response alignment can change.** In Figure 5a, we show an example where the true response alignment is safe during the first 30% of CoT, but shifts to unsafe in later CoT sentences. To accurately predict alignment while the final response changes, the predictor must process the evolving CoT instead of relying only on the prompt. The prompt would yield a static prediction that cannot capture this dynamic shift.
>
> > **W2**: This invalidates the comparison with text based monitoring, since the text of the CoT by itself does not also include the prompt.
>
> To clarify, for text-based monitoring methods, we do include the prompt in our setup to create a parallel comparison with activation-based monitoring.
> As shown in Appendix A.2.1 (Line 918), the instructions provided to the LLM monitor explicitly include both the prompt and the CoT:
> ```
> Task Prompt: You are given a harmful prompt question and a model’s chains of thoughts (CoTs).
>
> Prompt: {input_prompt}
>
> CoTs: {CoT}
> ```
> Similarly, for the fine-tuned BERT classifier and harmfulness classifiers, the input consists of the concatenation of the prompt and the CoT text. Therefore, both activation-based and text-based methods have access to the same information (prompt + CoT). The superior performance of the linear probe is not due to having access to additional information.
>
> >**Q1**: In Line 193, what is the value of d? That is, how aggressive is the dimensionality reduction with PCA?
>
> The original dimensionality d depends on the base model architecture:
> - s1.1-7B and R1-Distill-Qwen-7B (based on Qwen-2.5-7B): d = 3584
> - s1.1-14B and s1.1-32B: d = 53120
> - R1-Distill-Llama-8B: d = 4096
>
> We experimented with different PCA dimensions ∈ {32, 50, 64, 100, 128, 256, d} and found that 50 consistently gives the best overall performance across models and datasets. Performing dimensionality reduction improves computational efficiency, and also helps prevent overfitting, particularly in low-data regimes (as shown in Figure 2).
>
> >**Q2**: How is “foresight” used for training? As per the task description, only “observed” must be used to predict eventual misalignment.
>
> To clarify, “foresight” determines the labels (which response we take, and whether it is safe or unsafe) used during training, while “observed” determines the input features (CoT activations). Suppose we train a probe with 20 observed and 10 foresight. The training inputs are activations $h_{20}$ collected after the first 20 CoT sentences, and the labels are the alignment of response $A_{30}$ generated after 30 CoT sentences (20 + 10). This tests whether we can train a predictor that can “skip ahead” and predict future alignment outcomes.
>
> > **Q3**: It is surprising to see that the prediction accuracy is already very high with even 0-10% CoT (Figure 4). How is that the case? It would be helpful if the authors can provide a couple of examples of such cases. Is this a by-product of the prompt being encoded in activations used as input to the linear probe?
>
> We respectfully disagree that prediction performance is “very high” at 0-10% CoT. As shown in Figure 4, performance at 0% CoT (prompt only) is ~50 F1, which is the lowest point and close to the baseline. Performance gradually increases as more CoT is observed, reaching ~75-80 at 100% CoT. This corroborates our response to W1: while the prompt provides a weak baseline signal, the CoT itself is necessary for reliable prediction.
>
> However, we acknowledge that in a subset of cases, the prompt or first few CoT sentences can provide early signals. For instance, when models are already starting to comply and provide information in the first CoT sentence (e.g., “Okay, the user asks about … and here’s how we can achieve it.”), the CoT and answer stay unsafe. The key insight of our work is that in many cases, these early CoT sentences are **not** predictive of future responses; otherwise, we would have obtained a flat line in Figure 4.

---

### Meta-Review · Area_Chair_ZGbs · 2026-01-01

**Summary:**

This work adopts CoT activations to predict whether the response will be safe or unsafe for large reasoning models.

Most reviewers recognized that the paper is well written and easy to read, the experiments are extensive.
There are different opinions about the task and the finding, including its motivation and novelty. However, I think that the authors have provided relatively good clarification about these concerns.

One of the critical points is the lack of technical details, such as the design that "train probes on the last layer of the last token position". The authors didn't provide clear explanations.

There are also several concerns about the experiments, which have not been fully addressed in the rebuttal.

I think the most important reason of the low scores is that most reviewers didn't consider this work could make significant contributions to the community, methodology or insight.

Thus, the recommendation is reject.

**Reviewer Concerns:**

please refer to the summary

**Reviewer Scores:**

please refer to the summary

---

### Decision · Program_Chairs · 2026-01-26

Reject